# Stochastic Online Conformal Prediction with Semi-Bandit Feedback

## Abstract

Conformal prediction has emerged as an effective strategy for uncertainty quantification by modifying a model to output sets of labels instead of a single label. These prediction sets come with the guarantee that they contain the true label with high probability. However, conformal prediction typically requires a large calibration dataset of i.i.d. examples. We consider the online learning setting, where examples arrive over time, and the goal is to construct prediction sets dynamically. Departing from existing work, we assume semi-bandit feedback, where we *only observe the true label if it is contained in the prediction set*. For instance, consider calibrating a document retrieval model to a new domain; in this setting, a user would only be able to provide the true label if the target document is in the prediction set of retrieved documents. We propose a novel conformal prediction algorithm targeted at this setting, and prove that it obtains sublinear regret compared to the optimal conformal predictor. We evaluate our algorithm on a retrieval task, an image classification task, and an auction price-setting task, and demonstrate that it empirically achieves good performance compared to several baselines.

## 1 Introduction

Uncertainty quantification is an effective strategy for improving trustworthiness of machine learning models by providing users with a measure of confidence of each prediction to better inform their decisions. Conformal prediction (Vovk et al., 2005; Tibshirani et al., 2019; Park et al., 2019; Bates et al., 2021) has emerged as a promising strategy for uncertainty quantification due to its ability to provide theoretical guarantees for arbitrary blackbox models. They modify a given blackbox model $f : \mathcal{X} \to \mathcal{Y}$ to a conformal predictor $C : \mathcal{X} \to 2^{\mathcal{Y}}$ that predicts sets of labels. In the batch setting, it does so by using a held-out *calibration dataset* to assess the accuracy of $f$; it constructs smaller prediction sets if $f$ is more accurate and larger ones otherwise. Then, conformal prediction guarantees that the true label is contained in the prediction set with high probability—i.e.,

$$\mathbb{P}[y^* \in C(x)] \geq \alpha,$$

where $\alpha$ is the desired coverage rate, and the probability is taken over the random sample $(x, y^*) \sim D$ and the calibration dataset $Z = \{(x_i, y_i)\}_{i=1}^n \sim D^n$.

Traditional conformal prediction require the calibration set to consist of i.i.d. or exchangeable samples from the target distribution; furthermore, the calibration set may need to be large to obtain good performance. In many settings, such labeled data may not be easy to obtain. As a motivating setting, we consider a document retrieval problem, where the input $x$ might be a question and the goal is to retrieve a passage $y$ from a knowledge base such as Wikipedia that can be used to answer $x$; this strategy is known as retrieval-augmented question answering, and can mitigate issues such as hallucinations (Lewis et al., 2020; Shuster et al., 2021; Ji et al., 2023). A common practice is to use a retrieval model trained on a large dataset in a different domain in a zero-shot manner; for instance, a user might use the dense passage retrieval (DPR) model (Karpukhin et al., 2020) directly on their own dataset without finetuning. Thus, labeled data from the target domain may not be available.

In this paper, we propose an online conformal prediction algorithm that sequentially constructs prediction sets $C_t$ as inputs $x_t$ are given. We consider semi-bandit feedback, where we only observe the true label $y_t^*$ if it is included in our prediction set (i.e., $y_t^* \in C_t$). Otherwise, we observe an indicator that $y_t^* \notin C_t$, but do not observe $y_t^*$. In the document retrieval example, users might

sequentially provide queries $x_t$ to the DPR model, and our algorithm responds with a prediction set $C_t$ of potentially relevant documents. Then, the user selects the ground truth document $y_t^*$ if it appears in this set, and indicates that their query was unsuccessful otherwise.

As in conformal prediction, we aim to ensure that we achieve the desired coverage rate $\alpha$ with high probability. Letting $C_t^*$ denote the optimal prediction sets (i.e., the prediction sets given infinite calibration data), our algorithm ensures that with high probability, $C_t^* \subseteq C_t$ on all time steps $t$. Since $C_t^*$ achieves the desired coverage rate with high probability, this property ensures $C_t$ does so as well.

A trivial solution is to always include all documents in the prediction set. However, this would be unhelpful for the user, who needs to manually examine all documents to identify the ground truth one. Thus, an additional goal is to minimize the prediction set size. Formally, we consider the optimal prediction set $C_t^*$ in the limit of infinite data, and consider a loss function encoding how much worse our prediction set $C_t$ is compared to $C_t^*$. Then, our algorithm ensures this loss goes to zero as sufficiently many samples become available—i.e., the prediction sets $C_t$ converge to the optimal ones $C_t^*$ over time. Formally, our algorithm guarantees sublinear regret of $\tilde{\mathcal{O}}(\sqrt{T})$.

Finally, we empirically evaluate our algorithm on three tasks: image classification, document retrieval, and setting reservation prices in auctions. Our experiments demonstrate that our algorithm generates prediction sets that converge to the optimal ones while maintaining the desired coverage rate. Moreover, our algorithm significantly outperforms three natural baselines; each baseline either achieves worse cumulative expected regret or does not satisfy the desired coverage rate.

**Contributions.** We formalize and solve the problem of online conformal prediction with semi-bandit feedback. Our algorithm constructs compact prediction sets, ensuring high coverage probability. The algorithm also provides an efficient method for collecting large datasets that are expensive to label. Instead of asking the user to select the ground truth label from the set of all candidate labels, our approach only requires the user to select from a subset. This efficiency gain can be substantial when the label space is large. We assess the algorithm's performance on image classification, document retrieval and reservation price-setting in auctions, showcasing its effectiveness in a wide variety of real world applications, which highlights the practical utility of our approach.

**Related work.** Recent work has studied conformal prediction in the online setting (Gibbs and Candes, 2021; Bastani et al., 2022; Gibbs and Candès, 2022; Angelopoulos et al., 2024a;b). These approaches are motivated by conformal prediction for time series data, where the labels can shift in complex and potentially adversarial ways. As a consequence, they make very different assumptions. In particular, they allow for adversarial rather than i.i.d. assumptions; however, almost all of them assume that the ground truth label is observed at every step, regardless of whether it is contained in the prediction set. For example, the ACI algorithm proposed in Gibbs and Candes (2021) requires observing $y_t^*$ at every step since it is needed to update the quantile function $\hat{Q}_t(\cdot)$, which is needed to compute the prediction set $\hat{C}_t(\alpha_t)$ as well as the loss and gradient update. As the authors point out, they do not need to update the quantile function at every step. However, the steps where they update $\hat{Q}_t$ cannot be chosen in a way that depends on $y_t^*$, since doing so would lead to a biased estimate of the quantile function. In our experiments, we demonstrate that in the semi-bandit feedback setting, updating $\hat{Q}_t$ when $y_t^*$ is in the prediction set can create such a bias that leads ACI to fail to achieve coverage.

Similarly, the SAOCP algorithm proposed by Bhatnagar et al. (2023) requires observing $y_t^*$ at every step to construct the prediction set $S_t = \inf\{s \in R : y_t^* \in \hat{C}_t(X_t, s)\}$, which is then used to compute the loss and gradient, and the Conformal PID control algorithm proposed by Angelopoulos et al. (2024a) uses $y_t^*$ to compute the score $s_t = s_t(x_t, y_t^*)$, which is needed to compute the gradient. In contrast, we are motivated by the active learning setting, where users are interested in constructing conformal predictors on-the-fly rather than providing a calibration set ahead-of-time. Thus, in our setting, it is reasonable to assume that the data arrives i.i.d.; however, we need to handle semi-bandit feedback since the user may be unable to provide the true label $y_t^*$ if it is not in the prediction set.

Recently, Angelopoulos et al. (2024b) has pointed out that ACI still works if we take the "quantile function" to be the identity function—i.e., $\hat{Q}_t(\alpha) = \alpha$. This strategy achieves the desired coverage rate without requiring observing the ground truth labels $y_t^*$; instead, they only need to observe whether $y_t^*$ is contained in the prediction set—i.e., $\mathbb{1}(y_t^* \in \hat{C}_t)$. However, because they forego estimating the quantile function, the resulting algorithm is extremely sensitive to the structure of the scoring function as well as their choice of hyperparameters (in particular, the learning rate). In our experiments, we

show that a standard variation the scoring function (in particular, using logits instead of prediction probabilities) causes their performance to significantly degrade.

Angelopoulos et al. (2024b) also point out an additional shortcoming of ACI, which is that it does not guarantee that prediction sets converge to the "optimal" ones in the i.i.d. setting. As noted by Bastani et al. (2022), this issue is reflected in the fact that achieving $\alpha$ coverage (the guarantee satisfied by the ACI algorithm) can be achieved by the "cheating" strategy that outputs $\hat{C}_t = \mathcal{Y}$ with probability $1 - \alpha$ and $\hat{C}_t = \varnothing$ with probability $\alpha$. To remedy this issue, Angelopoulos et al. (2024b) proposes a modification to their algorithm that is guaranteed to converge to the optimal threshold in the setting of i.i.d. observations. However, their result is asymptotic (in addition to suffering from the sensitivity of the scoring function discussed above). Under certain assumptions, we prove that our algorithm satisfies a much stronger regret guarantee; to the best of our knowledge, this is the first regret guarantee on the performance of the prediction sets in terms of convergence of $\tau_t$ to $\tau^*$.

An additional advantage of our algorithm is that it guarantees that $\tau_t \leq \tau^*$ with high probability. This guarantee mirrors the distinction between marginal guarantees (Vovk et al., 2005) and probably approximately correct (PAC) (or training conditional) guarantees in the batch setting (Vovk, 2012; Park et al., 2019). Marginal guarantees have the form $\mathbb{P}_{Z \sim D^n, (x, y^*) \sim D}[y^* \in C_Z(x)] \geq \alpha$, i.e., $\alpha$ coverage over randomness in both the calibration set $Z$ and the new example $(x, y^*)$. They have the additional advantage that coverage converges to $\alpha$ with the number of calibration examples. In contrast, PAC guarantees disentangle these two sources of randomness:

$$\mathbb{P}_{Z \sim D^n}[\mathbb{P}_{(x, y^*) \sim D}[y^* \in C_Z(x)] \geq \alpha] \geq 1 - \delta,$$

for a given $\delta \in \mathbb{R}_{>0}$. In other words, coverage holds with high probability over $Z$. Our guarantee that $\tau_t \leq \tau^*$ is equivalent to providing the coverage guarantee $\mathbb{P}_{(x, y^*) \sim D}[y^* \in C_{\tau_t}(x)] \geq \alpha$ for *every* $t$ with probability at least $1 - \delta$ over the whole time horizon. To the best of our knowledge, existing online conformal prediction algorithms all provide a marginal coverage guarantee (it is not even clear how a PAC guarantee would look in the adversarial setting since the calibration examples are not random). For the active learning setting, a PAC guarantee makes sense since it ensures that our algorithm satisfies the desired coverage rate for every user-provided input. Finally, while we do not provide an explicit guarantee that the coverage rate converges to $\alpha$, our regret bound ensures that $\tau_t \to \tau^*$, which effectively ensures convergence to an $\alpha$ coverage rate.

## 2 PROBLEM FORMULATION

Let $\mathcal{X}$ denote the inputs and $\mathcal{Y}$ denote the labels. Let $[T] = \{1, 2, \cdots, T\}$, and let $t \in [T]$ be the steps on which examples $(x_t, y_t^*)$ arrive, where $x_t$ is the input on step $t$ and $y_t^*$ is the corresponding ground truth label. We assume given a scoring function $f : \mathcal{X} \times \mathcal{Y} \mapsto \mathbb{R}$ (also called the *non-conformity score*). The scoring function captures the confidence in whether $y$ is the ground truth label for $x$.

We consider a fixed distribution $D$ over $\mathcal{X} \times \mathcal{Y}$; let $\mathbb{P}(x, y)$ be the corresponding probability measure. On each step $t$, a sample $(x_t, y_t^*) \sim D$ is drawn. Then, our algorithm observes $x_t$, and constructs a prediction set $C_t \subseteq \mathcal{Y}$ of form

$$C_t = \{y \in \mathcal{Y} \mid f(x_t, y) > \tau_t\},$$

where $\tau_t \in \mathbb{R}$ is a parameter to be chosen. In other words, our prediction set $C_t$ include all labels with score at least $\tau_t$ in round $t$. Note that a smaller (resp., larger) $\tau_t$ corresponds to a larger (resp., smaller) prediction set $C_t$. Then, our algorithm receives *semi-bandit feedback*—i.e., it only observes $y_t^*$ if $y_t^* \in C_t$. If $y_t^* \notin C_t$, it receives feedback in the form of a binary indicator that $y_t^* \notin C_t$.

Next, we describe our desired correctness properties. First, given a coverage rate $\alpha \in [0, 1)$, our goal is to ensure that we cover the ground truth label with probability at least $\alpha$ on all time steps:

$$\forall t \in [T] . \mathbb{P}[y_t^* \in C_t] \geq \alpha. \tag{1}$$

We want (1) to hold with high probability. A trivial solution to the problem described so far is to always take $\tau_t = -\infty$. Thus, we additionally want to minimize some measure of the prediction set size. We consider a loss function $\phi : \mathbb{R} \to \mathbb{R}$, where $\phi(\tau_t)$ encodes the loss incurred on step $t$. Then, our goal is to converge to the best possible prediction sets over time, which we formalize by aiming to achieve sublinear regret. To this end, consider the optimal prediction set $C_t^* \subseteq \mathcal{Y}$ defined by

$$C_t^* = \{y \in \mathcal{Y} \mid f(x_t, y) > \tau^*\},$$

where $\tau^*$ is defined by

$$\tau^* = \arg\max_{\tau \in \mathbb{R}} \tau \quad \text{subj. to} \quad \mathbb{P}[f(x_t, y_t^*) \geq \tau] \geq \alpha.$$

Note that $f(x_t, y_t^*) \geq \tau$ iff $y_t^* \in C_t^*$, so this property says that $\mathbb{P}[y_t^* \in C_t^*] \geq \alpha$. By definition, $C_t^*$ is the smallest possible prediction set for $x_t$ that achieves a coverage rate of $\alpha$. Then, the expected cumulative regret is

$$R_T = \mathbb{E}\left[\sum_{t \in [T]} \phi(\tau_t) - \phi(\tau^*)\right].$$

For technical reasons, we need to impose an assumption on our loss. Intuitively, if the loss $\phi$ is discontinuous at $\tau^*$, then we may achieve linear regret since $\phi(\tau_t) - \phi(\tau^*) \geq c > 0$ for some constant $c$. A natural way to formalize the continuity assumption is based on the cumulative distribution function (CDF) of the scoring function. In particular, let $G^*$ be the CDF of the random variable $s = f(x, y^*)$, where $(x, y^*) \sim D$. Then, we have the following assumption:

**Assumption 2.1.** $\phi(\tau) = \psi(G^*(\tau))$ for some $K$ Lipschitz continuous function $\psi$ (with $K \in \mathbb{R}_{>0}$).

That is, if $G^*(\tau)$ is very flat in some region, then $\phi(\tau)$ cannot vary very much in that region. Intuitively, if $G^*(\tau)$ is flat in some region, then we obtain very few samples in that region, so it becomes very hard to estimate $\tau$ in that region. If $\phi(\tau)$ varies a lot in this region, then the regret can be large if $\tau^*$ lies in that region. One caveat is this assumption precludes prediction set size as a loss, since prediction set size is discontinuous. Next, we assume our loss is bounded:

**Assumption 2.2.** $\phi_{max} := \|\phi\|_\infty < \infty$.

Intuitively, Assumption 2.2 says that the overall reward is bounded, so we cannot accrue huge regret early on in our algorithm when we have very little data. Finally, we make the following assumption:

**Assumption 2.3.** $G^*(\tau^*) = 1 - \alpha$.

Note that $G^*(\tau)$ is the miscoverage rate of our algorithm for parameter $\tau$. Thus, this assumption says the optimal parameter value $\tau^*$ achieves a coverage rate of exactly $\alpha$, which simplifies our analysis.

Then, our goal is to find the optimal prediction sets $C_t^*$ with coverage rate $\alpha$. Intuitively, $C_t^*$ is the smallest set that contains the ground truth label with a high probability. At each step, the algorithm observes $x_t$ and returns a set $C_t$ of candidate labels, and the user either (1) selects the ground truth label $y_t^*$ from $C_t$, or (2) indicates that the ground truth label is not in $C_t$. In a document retrieval setting, $x_t$ is a query sent by the user, and $y_t^*$ is the ground truth document, while in an image classification setting, $x_t$ is an image and $y_t^*$ is the ground truth class.

## 3 ALGORITHM

Next, we describe our online conformal prediction algorithm (summarized in Algorithm 1). As before, let $s = f(x, y^*)$ be the random variable that is the score of a random sample $(x, y^*) \sim D$, and let $G^*$ be its CDF. By definition, $G^*(\tau) = \mathbb{P}[f(x, y^*) \leq \tau]$ is the miscoverage rate, so

$$\tau^* = \sup\{\tau : G^*(\tau) \leq 1 - \alpha\}.$$

Thus, if we know $G^*$, then our problem can be solved by choosing $\tau_t = \tau^*$ for all $t$. However, since we do not know $G^*$, we can solve the problem by estimating it from samples. Denote the estimated CDF after step $t$ as $G_t$. A naïve solution is to choose

$$\tilde{\tau}_t = \sup\{\tau \in \mathbb{R} \mid G_{t-1}(\tau) \leq 1 - \alpha\}.$$

However, this strategy may fail to satisfy our desired coverage rate due to randomness in our estimate $G_{t-1}$ of $G^*$. Failing to account for miscovered examples can exacerbate this problem—if we ignore samples where we failed to cover $y_t^*$, then our estimate $G_t$ becomes worse, thereby increasing the chance that we will continue to fail to cover $y_t^*$. This feedback loop can lead to linear regret.

To address this challenge, we instead use a high-probability upper bound on $G^*$. In particular, for a error bound $\delta \in \mathbb{R}_{>0}$ to be specified, we construct a $1 - \delta$ confidence bound for the empirical CDF

---

**Algorithm 1** Semi-bandit Prediction Set (SPS)

---

**Input:** horizon $T$, desired quantile $\alpha$
$\tau_1 \leftarrow -\infty$
**for** $t = 1$ **to** $T$ **do**
   **if** $s_t \geq \tau_t$ **then**
      observe $s_t$
   **else**
      $s_t \leftarrow \tau_t$
   **end if**
   Compute $\overline{G}_t$ according to (4)
   $\tau_{1-\alpha,t} \leftarrow \sup\{\tau \in \mathbb{R} \mid \overline{G}_t(\tau) \leq 1 - \alpha\}$
   $\tau_t \leftarrow \max\{\tau_{1-\alpha,t}, \tau_t\}$
**end for**

---

$G_t$ using the Dvoretzky–Kiefer–Wolfowitz (DKW) inequality (Massart, 1990). Letting $\overline{G}_t$ be the upper confidence bound, we instead aim to choose

$$\tau_t = \sup\{\tau \in \mathbb{R} \mid \overline{G}_{t-1}(\tau) \leq 1 - \alpha\}.$$

On the event that $\overline{G}_{t-1}$ is a valid upper bound, then we have $\tau_t \leq \tau^*$. This property ensures that we always cover the ground truth label, which ensures that our subsequent CDF estimate $\overline{G}_t$ is valid. As a consequence, our algorithm converges to the true $\tau^*$.

One remaining issue is how to handle steps where $y_t^* \notin C_t$. On these steps, our algorithm substitutes $\tau_t$ for the observation $f(x_t, y_t^*)$. Intuitively, the reason this strategy works is that the learner does not need to accurately estimate $G^*$ in the interval $[0, \tau^*)$ to recover $\tau^*$; it is sufficient to include the right fraction of samples in this interval. As long as our algorithm maintains the property that $\tau_t \leq \tau^*$, then $\tau_t$ lies in this interval, so substituting $\tau_t$ is sufficient.

Relatedly, our algorithm includes a constraint that $\tau_{t+1} \geq \tau_t$ for all $t$. We include this constraint because our estimate $G_t$ of the CDF in the interval $[0, \tau_t)$ may be flawed due to semi-bandit feedback. By avoiding going backwards, we ensure that these flaws do not affect our choice of $\tau_{t+1}$. Again, as long as $\tau_t \leq \tau^*$, this constraint does not prevent convergence.

Now, we formally define $\overline{G}_t$ as follows. First, define the truncated CDF $G_t^*(\tau)$ by

$$G_t^*(\tau) = \begin{cases} 0 & \text{if } \tau < \tau_t \\ G^*(\tau) & \text{if } \tau \geq \tau_t. \end{cases} \tag{2}$$

This CDF captures the CDF where we replace samples $s \leq \tau_t$ with $\tau_t$. In particular, $G_t^*(\tau)$ shifts all the probability mass of $G^*$ in the region $[0, \tau_t]$ to a point mass at $\tau_t$. Next, the corresponding empirical CDF $G_t$ is

$$G_t(\tau) = \frac{1}{t} \cdot \sum_{j=1}^{t} \mathbb{1}(\max\{\tau_t, s_j\} \leq \tau). \tag{3}$$

Finally, letting $\delta = 2/T^2$ and $\epsilon_t = \sqrt{\log(2/\delta)/2t}$, the upper bound $\overline{G}_t$ on $G_t$ from DKW is

$$\overline{G}_t(\tau) = G_t(\tau) + \epsilon_t. \tag{4}$$

We use this $\overline{G}_t$ to compute $\tau_t$ in Algorithm 1.

Finally, Algorithm 1 satisfies the following theoretical guarantee: (i) it incurs sublinear regret $\tilde{\mathcal{O}}(\sqrt{T})$, and (ii) it satisfies $C_t^* \subseteq C_t$ for all $t$ with probability at least $1 - 2/T$ (see Appendix A for a proof):

**Theorem 3.1.** *The expected cumulative regret of Algorithm 1 satisfies*

$$R_T \leq K\left(2\log T + 4\sqrt{T \log T} + 1\right) + 4\phi_{max},$$

*In addition, with probability at least $1 - 2/T$, we have $\tau_t \leq \tau^*$ for all $t$.*

## 4 EXPERIMENTS

### 4.1 EXPERIMENTAL SETUP

**Image classification task.** We use the Vision Transformer (Dosovitskiy et al., 2020) model on the ImageNet dataset (Deng et al., 2009).[1] Each image from the dataset belongs to exactly one class out of the 1,000 candidate classes. Consequently, the cardinality of the label domain is 1,000 (i.e., $|\mathcal{Y}| = 1000$). In the experiments, images arrive sequentially, and our algorithm aims to construct the smallest candidate label set that achieves the desired coverage. We use the logits score returned by a ViT model (pretrained on the ImageNet training set) as our scoring function $f$; we use the logits instead of the softmax function to evaluate sensitivity to the choice of scoring function.

**Document retrieval task.** Next, we consider the Dense Passage Retriever (DPR) model (Karpukhin et al., 2020) on the SQuAD question-answering dataset. DPR leverages a dual-encoder architecture that maps questions and candidate documents to embedding vectors. Denoting the space of questions and documents by $\mathcal{Q}$ and $\mathcal{D}$, respectively, then DPR consists of a question encoder $E_Q : \mathcal{Q} \mapsto \mathbb{R}^{768}$ and a document encoder $E_D : \mathcal{D} \mapsto \mathbb{R}^{768}$. Given a question $q$ and a set of candidate documents $D \subseteq \mathcal{D}$, the similarity score between each document $d \in D$ and the question $q$ is defined as:

$$s_{q,d} = \frac{E_Q(q) \cdot E_D(d)}{|E_Q(q)| \cdot |E_D(d)|}.$$

We use this score as our scoring function. Then, our goal is to construct the smallest set of candidate documents while guaranteeing that they contain the ground truth document with high probability.

Our dataset is SQuAD question-answering dataset (Rajpurkar et al., 2016), which is a popular benchmark for reading comprehension. Each question in SQuAD can be answered by finding the relevant information in a corresponding Wikipedia paragraph known as the context. The authors of DPR make a few additional changes to adapt SQuAD better for document retrieval. First, paragraphs are further split into multiple, disjoint text blocks of 100 words, serving as the basic retrieval unit (i.e., candidate documents). Second, each question is paired with ground truth documents and a set of irrelevant documents.[2] In our experiments, for each question, we include one ground truth document and all the irrelevant documents to create the set of candidate documents.

**Second-price auctions task.** Lastly, we consider the scenario of setting reservation prices in second-price auctions, a well-studied problem that has semi-bandit feedback (Cesa-Bianchi et al., 2014; Zhao and Chen, 2020). In this problem, a seller (the auctioneer) repeatedly sells the same type of items to a group of bidders. In each round $t$, she publicly announces a reservation price $p^t$, while bidders draw their private values $v^t$ from a fixed distribution that is unknown to the seller. For each bidder $i$, she will submit the bid $B_i^t = v_i^t$ if and only if $v_i^t \geq p^t$. The seller obtains reward $R_t$:

$$R_t = \begin{cases} 0 & \text{if } p^t > B_t^{(1)} \\ p^t & \text{if } B_t^{(2)} < p^t \leq B_t^{(1)} \\ B_t^{(2)} & \text{if } p^t \leq B_t^{(2)}, \end{cases}$$

where $B_t^{(1)}, B_t^{(2)}$ denote the highest and second-highest bid received by the seller in round $t$, implying that the seller only observes bids that are higher than $p^t$. We consider a seller that aims to learn

$$p^* = \arg\max_{p \in \mathbb{R}} p \quad \text{subj. to} \quad \mathbb{P}[B^{(1)} \geq p] \geq \alpha.$$

That is, the seller aims to find the highest reservation price $p$ such that she can sell the item with probability at least $\alpha$. This problem can be solved using online conformal prediction. Following standard practice (Mohri and Medina, 2014), we use a synthetic dataset that adapts from eBay auction data (Jank and Shmueli, 2010). Specifically, we simulate the distribution of $v_i$ by using the empirical distribution of the observed bids in the dataset.

---

[1] Obtained from https://www.image-net.org/ with a custom and non-commercial license; we use the $16 \times 16$ down sampled version.

[2] In the data, all questions are paired with 50 irrelevant documents. We construct the candidate documents set by including all 50 irrelevant documents and 1 ground truth document. We exclude questions that have 0 ground truth documents. The data were obtained from DPR's public Github Repo: https://github.com/facebookresearch/DPR with licenses CC BY-SA 4.0 and CC BY-NC 4.0

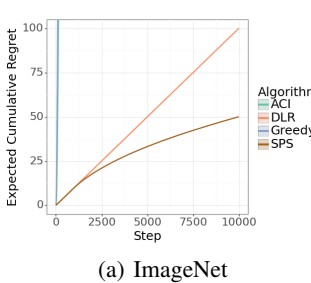 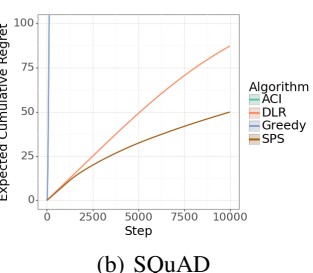 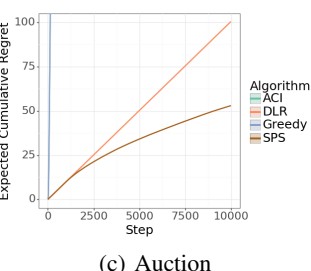

(a) ImageNet      (b) SQuAD      (c) Auction

Figure 1: **Cumulative Regret**

**Baselines.** We compare to greedy, Adaptive Conformal Inference (ACI) (Gibbs and Candes, 2021), and Decaying Learning Rate (DLR) (Angelopoulos et al., 2024b). The greedy strategy chooses

$$\tau_t = \sup\{\tau \in \mathbb{R} \mid G_t(\tau) \leq 1 - \alpha\}$$

at every step, which cannot guarantee the $\alpha$ coverage rate, leading it to undercover significantly more than desired. Next, Adaptive Conformal Inference (ACI) adjusts $\alpha_t$ (and then $\tau_t$) based on whether the ground truth label is in the previous round's prediction set. We choose the learning rate $\gamma$ from a grid search in a candidate set proposed in (Gibbs and Candès, 2022). To run ACI in our semi-bandit feedback setting, we only update the quantile function $\hat{Q}_t$ when ground-truth label $y_t^*$ is observed; as we show, this biased strategy for updating $\hat{Q}_t$ leads it to fail to achieve the desired coverage rate.

Lastly, we consider the Decaying Learning Rate (DLR) algorithm proposed in (Angelopoulos et al., 2024b). In contrast to ACI, DLR directly performs gradient descent on the cutoffs $\tau_t$ instead of the quantiles $\alpha_t$; it can be viewed as running ACI with $\hat{Q}_t(\alpha) = \alpha$. We set the learning rate to the one used in the experiments from the original paper—i.e., $\eta_t = t^{-1/2-\epsilon}$ with $\epsilon = 0.1$. We show results for two additional baselines, explore-then-commit (ETC) and conservative ETC, in Appendix B.

**Experiment parameters.** We use $\alpha = 0.9$ and $T = 10000$, and report the averages across 5 runs.

**Metrics.** First, we consider the cumulative regret for the following reward function $\phi$:

$$\phi(\tau) = \begin{cases} -\lambda_1 \cdot |G^*(\tau) - (1-\alpha)| & \text{if } G^*(\tau) \leq (1-\alpha) \\ -\lambda_2 \cdot |G^*(\tau) - (1-\alpha)| & \text{if } G^*(\tau) > (1-\alpha), \end{cases}$$

for some $0 < \lambda_1 < \lambda_2$; we take $\lambda_1 = 0.1$ and $\lambda_2 = 10$. Note this loss imposes a larger penalty for undercovering compared to overcovering. Next, we consider coverage rate:

$$\text{Coverage Rate} = \frac{1}{T} \sum_{t=1}^{T} \mathbb{1}(y_t^* \in C_t).$$

Third, we consider the number of times $\tau_t > \tau^*$, which measures the violation of our safety condition:

$$\text{Undercoverage Count} = \sum_{t=1}^{T} \mathbb{1}(\tau_t > \tau^*).$$

Our goals are to (i) achieve $\tilde{\mathcal{O}}(\sqrt{T})$ regret, while (ii) maintaining the desired $\alpha$ coverage rate, and (iii) achieving zero undercoverage count with probability at least $1 - \delta$ over the entire time horizon.

### 4.2 RESULTS

**Regret.** First, Figure 1 shows the cumulative regret of each approach on each task. As can be seen, our algorithm consistently obtains the lowest regret. The ACI algorithm attains a regret level comparable to the greedy algorithm because its quantile function is updated with a bias. Furthermore, note that the curves for both ACI and the greedy algorithm appear to be superlinear. This can happen since these algorithms do not properly account for semi-bandit feedback—in particular, the empirical estimate $G_t$ of the distribution becomes increasingly truncated.

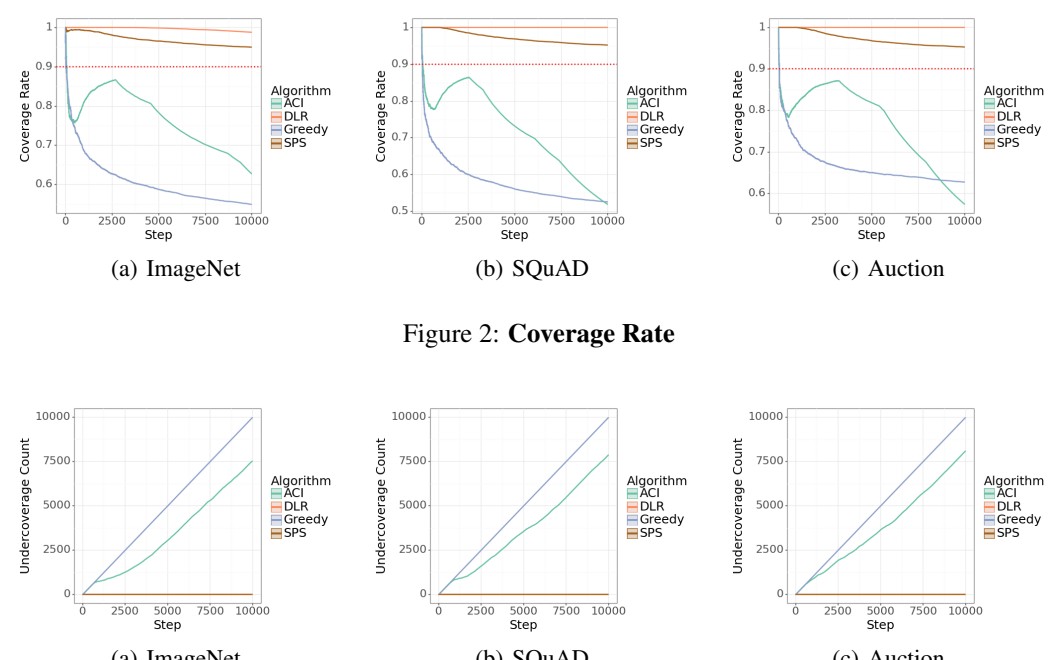

Figure 2: **Coverage Rate**

Figure 3: **Undercoverage Count**

Moreover, the poor performance of DLR can be attributed to the fact that it does not make use of a quantile function. As a consequence, without score-specific hyperparameter tuning (i.e., tuning the learning rate), it can converge very slowly to the optimal prediction sets. For many scoring functions, we do not have prior knowledge of the score's range, which exacerbates these issues. These issues are particularly salient when we consider tasks such as the second-price auction, where the score's range (i.e., the range of bids) can be difficult to predict in advance. In contrast, our algorithm consistently performs and does not have any hyperparamters to tune.

**Coverage rate.** Next, Figure 2 shows the coverage rate achieved by each algorithm for each task. Both ACI and greedy fail to maintain the desired coverage rate in all three tasks. DLR and SPS both achieve the desired coverage rate; however, SPS converges more quickly.

**Undercoverage count.** Finally, Figure 3 shows the undercoverage count. Note that greedy and ACI frequently undercover. ACI has high undercoverage count because the prediction set oscillates between being too small (i.e., $\tau_t > \tau^*$) and too large (i.e., $\tau_t < \tau^*$). This behavior can be undesirable since it means that different inputs have different coverage probabilities. In contrast, our algorithm never undercovers since $\tau_t$ is guaranteed to converge to $\tau^*$ from below. Interestingly, DLR also does not undercover. While their algorithm is not guaranteed to satisfy this property, it incrementally estimates $\tau_t$ by starting from a conservative $\tau$. Thus, if the learning rate is small enough, it would not undercover until $\tau_t$ gets significantly closer to the true $\tau^*$.

**Summary.** These results demonstrate that our algorithm achieves (and converges to) the desired coverage rate while achieving sublinear regret and maintaining $\tau_t < \tau^*$. In contrast, ACI and greedy fail to achieve the desired coverage rate, and DLR converges much more slowly than our algorithm.

## 5 CONCLUSION

We have proposed a novel conformal prediction algorithm for constructing online prediction sets under stochastic semi-bandit feedback. We have shown our our algorithm can be applied to learn optimal prediction sets in image classification, document retrieval, and second-price auction reservation price prediction. Our experiments demonstrate that we maintain the desired $\alpha$ coverage level while achieving prediction set sizes that achieve sublinear regret and zero undercoverage count.

**Ethics statement:** Our approach aims to improve the trustworthiness of machine learning models by reliably quantifying their uncertainty. We do not foresee any ethical concerns with our work.

**Reproducibility statement:** We provide detailed explanations of our experiments in Section 4.2 and include the necessary parameter settings in the captions of the result figures. All experiments were conducted on a MacBook with an M1 chip.

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

## A  PROOF OF THEOREM 3.1

We begin by proving several helper lemmas, and then establish our main result. Our first lemma shows that $G_t$ defined in (3) converges uniformly to $G_t^*$ defined in (2). Note that the randomness comes from the sequence of random variables $s_j = f(x_j, y_j^*)$.

**Lemma A.1.** *For any $\epsilon \in \mathbb{R}_{>0}$, we have*

$$\mathbb{P}\left[\sup_{\tau \in \mathbb{R}} |G_t(\tau) - G_t^*(\tau)| > \epsilon\right] \le 2e^{-2t\epsilon}.$$

*Proof.* Let the full empirical CDF of $s$ be $G_t'(\tau) = \frac{1}{t} \sum_{j=1}^t \mathbb{1}(s_j \le \tau)$, and define the events

$$A_< = \left\{\sup_{\tau < \tau_t} |G_t(\tau) - G_t^*(\tau)| > \epsilon\right\} \quad \text{and} \quad A_\ge = \left\{\sup_{\tau \ge \tau_t} |G_t(\tau) - G_t^*(\tau)| > \epsilon\right\}.$$

First, consider $A_\ge$. For $\tau \ge \tau_t$, by definition of $G_t^*$ and $G_t$, we have $G_t^*(\tau) = G^*(\tau)$ and $G_t(\tau) = G_t'(\tau)$. Thus, by the DKW inequality, we have $\mathbb{P}[A_\ge] \le 2e^{-2t\epsilon^2}$. Next, consider $A_<$. By definition of $G_t^*$ and $G_t$, we have $G_t(\tau) = G_t^*(\tau) = 0$, so $\mathbb{P}[A_<] = 0$. Thus, by a union bound, we have

$$\mathbb{P}\left[\sup_{\tau \in \mathbb{R}} |G_t(\tau) - G_t^*(\tau)| > \epsilon\right] \le \mathbb{P}[A_\ge] + \mathbb{P}[A_<] \le 2e^{-2t\epsilon^2},$$

as claimed. $\qquad \square$

Our next lemma shows that with high probability, the desired invariant $\tau_t \le \tau^*$ holds for all $t \in [T]$.

**Lemma A.2.** *Suppose $\sup_{\tau \in \mathbb{R}} |G_t(\tau) - G_t^*(\tau)| \le \epsilon_t$ for all $t \in [T]$. Then, $\tau_t \le \tau^*$ for all $t \in [T]$.*

*Proof.* We prove by induction. For the base case $t = 1$, we choose $\tau_1 = -\infty$, so $\tau_1 \le \tau^*$. For the inductive case, assume that $\tau_k \le \tau^*$. Now, by our assumption that $\sup_{\tau \in \mathbb{R}} |G_k(\tau) - G_k^*(\tau)| \le \epsilon_k$ for all $k \in [T]$, we have

$$G_{k-1}^*(\tau) \le G_{k-1}(\tau) + \epsilon_{k-1} = \overline{G}_{k-1}(\tau).$$

Because $\tau_k \le \tau^*$, we also have $G_{k-1}^*(\tau) = G^*(\tau)$ for all $\tau \ge \tau^*$. Together, we have

$$\overline{G}_{k-1}(\tau^*) \ge G^*(\tau^*) = 1 - \alpha.$$

Thus, we have $\tau_{1-\alpha,k} \le \tau^*$. Since $\tau_k \le \tau^*$, it follows that:

$$\tau_{k+1} = \max\{\tau_{k-1}, \tau_{1-\alpha,k}\} \le \tau^*,$$

as claimed. $\qquad \square$

Our next lemma bounds the range of $\overline{G}_t(\tau_t)$.

**Lemma A.3.** *Suppose $\sup_{\tau \in \mathbb{R}} |G_t(\tau) - G_t^*(\tau)| \le \epsilon_t$ for all $t \in [T]$. Then, for all $t \in [T]$, we have*

$$1 - \alpha - \frac{2}{t} \le \overline{G}_t(\tau_t) \le 1 - \alpha + 2\epsilon_t.$$

*Proof.* By Lemma A.2, $\tau_t \le \tau^*$, implying that $G^*(\tau_t) \le G^*(\tau^*)$. For the upper bound, we have

$$\overline{G}_t(\tau_t) \le G_t^*(\tau_t) + 2\epsilon_t = G^*(\tau_t) + 2\epsilon_t \le G^*(\tau^*) + 2\epsilon_t = 1 - \alpha + 2\epsilon_t.$$

Next, we consider the lower bound. When $t = 1$, the inequality trivially holds. Otherwise, at step $t - 1$, $\tau_t$ is at least the $\lfloor (1-\alpha)(t-1) \rfloor$-th order statistic. On the event that $s_t \ge \tau_t$, then $\tau_t$ is at least the $(\lfloor (1-\alpha)(t-1) \rfloor - 1)$-th order statistic. Thus, we have

$$\overline{G}_t(\tau_t) \ge \frac{\lfloor (1-\alpha)t \rfloor - 1}{t} \ge \frac{(1-\alpha)t}{t} - \frac{2}{t}.$$

If $s_t < \tau_t$, then $\tau_t$ is at least the $(\lfloor (1-\alpha)t \rfloor)$-th order statistic. Then, we have

$$\overline{G}_t(\tau_t) \ge \frac{\lfloor (1-\alpha)t \rfloor}{t} \ge \frac{(1-\alpha)t}{t} - \frac{2}{t},$$

as claimed. $\qquad \square$

**Theorem 3.1.** *The expected cumulative regret of Algorithm 1 satisfies*

$$R_T \leq K \left( 2 \log T + 4\sqrt{T \log T} + 1 \right) + 4\phi_{max},$$

*In addition, with probability at least $1 - 2/T$, we have $\tau_t \leq \tau^*$ for all $t$.*

*Proof.* Define the good event

$$E = \forall t \in [T] \ . \ \sup_{\tau \in \mathbb{R}} |G_t^*(\tau) - G_t(\tau)| \leq \epsilon_t.$$

By a union bound and by Lemma A.1, we have

$$\mathbb{P}[\neg E] \leq \sum_{t=1}^{T} 2e^{-2t\epsilon_t^2} \leq T\delta = \frac{2}{T}.$$

Now, we have

$$R_T = \mathbb{E}\left[ \sum_{t=1}^{T} |\phi(\tau^*) - \phi(\tau_t)| \ \Big| \ E \right] \cdot \mathbb{P}[E] + \mathbb{E}\left[ \sum_{t=1}^{T} |\phi(\tau^*) - \phi(\tau_t)| \ \Big| \ \neg E \right] \cdot \mathbb{P}[\neg E]$$

$$= \sum_{t=1}^{T} \mathbb{E}\left[ |\phi(\tau^*) - \phi(\tau_t)| \mid E \right] \cdot \mathbb{P}[E] + \sum_{t=1}^{T} \mathbb{E}\left[ |\phi(\tau^*) - \phi(\tau_t)| \mid \neg E \right] \cdot \mathbb{P}[\neg E]$$

$$\leq \sum_{t=1}^{T} \mathbb{E}\left[ |\phi(\tau^*) - \phi(\tau_t)| \mid E \right] \cdot \mathbb{P}[E] + \sum_{t=1}^{T} \mathbb{E}\left[ 2\phi_{max} \mid \neg E \right] \cdot \mathbb{P}[\neg E]$$

$$\leq \left( \sum_{t=1}^{T} \underbrace{\mathbb{E}\left[ |\phi(\tau^*) - \phi(\tau_t)| \mid E \right]}_{=:X_t} \right) + 4\phi_{max}.$$

The first inequality follows from Assumption 2.2. Using Lemma A.3, we can bound $X_t$ as follows:

$$X_t = \mathbb{E}\left[ |\phi(\tau^*) - \phi(\tau_t)| \mid E \right]$$
$$= \mathbb{E}[\psi(G^*(\tau^*)) - \psi(G^*(\tau_t)) \mid E]$$
$$\leq K \cdot \mathbb{E}[|G^*(\tau^*) - G^*(\tau_t)| \mid E]$$
$$= K \cdot \mathbb{E}\left[ |1 - \alpha - \overline{G}_t(\tau_t) + \overline{G}_t(\tau_t) - G^*(\tau_t)| \mid E \right]$$
$$\leq K \cdot \mathbb{E}\left[ |1 - \alpha - \overline{G}_t(\tau_t)| + |\overline{G}_t(\tau_t) - G^*(\tau_t)| \mid E \right]$$
$$\leq K \cdot \mathbb{E}\left[ \max\left\{ \frac{2}{t}, 2\epsilon_t \right\} + |\overline{G}_t(\tau_t) - G_t^*(\tau_t)| \ \Big| \ E \right]$$
$$\leq K \cdot \max\left\{ \frac{2}{t}, 2\epsilon_t \right\} + K \cdot \mathbb{E}\left[ \sup_{\tau \in \mathbb{R}} |G_t(\tau) - G_t^*(\tau)| \ \Big| \ E \right]$$
$$\leq K \cdot \max\left\{ \frac{2}{t}, 2\sqrt{\frac{\log(2/\delta)}{2t}} \right\} + 2K\sqrt{\frac{\log(2/\delta)}{2t}}$$
$$\leq \frac{2K}{t} + 4K\sqrt{\frac{\log(2/\delta)}{2t}},$$

where the first inequality follows from Assumption 2.1 and the third inequality follows from Lemma A.3. Thus, we have

$$\sum_{t=1}^{T} X_t \leq \sum_{t=1}^{T} \left\{ \frac{2K}{t} + 4K \cdot \sqrt{\frac{\log T}{t}} \right\} \leq K \left[ 2 \log T + 1 + 4\sqrt{T \log T} \right].$$

The claim follows. □

# B ADDITIONAL EXPERIMENTS

We consider two additional baselines. First, explore-then-commit (ETC) chooses $\tau_t = -\infty$ in the first $m$ steps, and then commits to

$$\tau_t = \sup\{\tau \in \mathbb{R} \mid G_m(\tau) \leq 1 - \alpha\}.$$

Next, conservative ETC (Con-ETC) uses the same strategy, except it commits to

$$\tau_t = \sup\{\tau \in \mathbb{R} \mid \overline{G}_m(\tau) \leq 1 - \alpha\}$$

after the exploration period. In other words, it commits to a conservative choice of $\tau$ that satisfies our coverage guarantee, and also guarantees $\tau_t \leq \tau^*$ with probability at least $1 - 2/T$. The number of exploration rounds are chosen via a grid search. Results are shown in Figures 4, 5, & 6. As can be seen, ETC fails to achieve the desired coverage rate since it does not account for uncertainty; conversely, Con-ETC achieves very high regret since it does not adaptively choose $\tau_t$ over time.

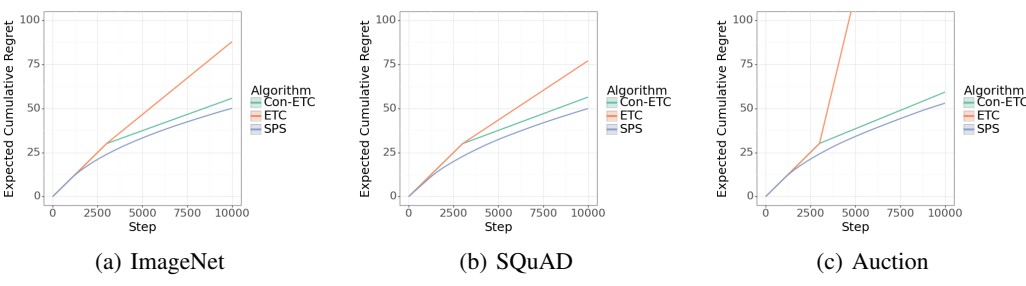

(a) ImageNet          (b) SQuAD          (c) Auction

Figure 4: **Cumulative Regret**

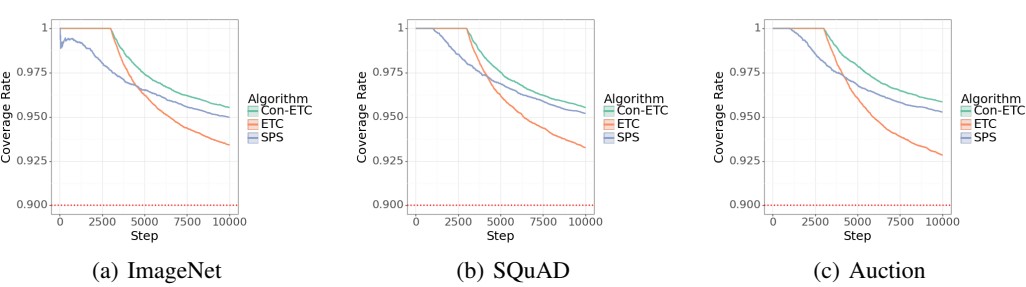

(a) ImageNet          (b) SQuAD          (c) Auction

Figure 5: **Coverage Rate**

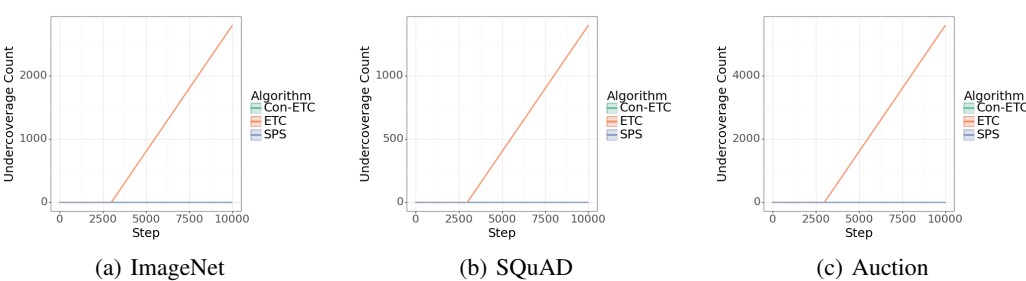

(a) ImageNet          (b) SQuAD          (c) Auction

Figure 6: **Undercoverage Count**