# OpenReview forum: "Stochastic Online Conformal Prediction with Semi-Bandit Feedback"
_ICLR.cc/2025/Conference — Submitted to ICLR 2025_

### Official Review · Reviewer_TNwk · 2024-10-20

**Soundness:** 2
**Presentation:** 2
**Contribution:** 2
**Rating:** 5
**Confidence:** 2

**Summary:**

The authors propose a novel conformal prediction algorithm for constructing online prediction sets under a novel setting called stochastic semi-bandit feedback. They prove that the developed algorithm can achieve sub-linear regret and demonstrate the applicability of their method in different tasks including a retrieval task, an image classification task, and an auction price-setting task.

**Strengths:**

1. Considered a novel setup where true label can be observed only when prediction set covers it

2. Established a regret bound for the problem of interest

3. Conducted experiments in different tasks demonstrating the utility of the proposed method

**Weaknesses:**

1. While authors established a regret bound for the problem of interest, the author didn't mention related regret bounds in the existing work. Besides, the author did not mention how good the regret bound is compared to the existing work.

2. In the experiments, I observe the coverage rate is relatively conservative for the proposed method. Does the coverage rate stabilize if step is large enough? Is it possible to speed up the coverage convergence?

**Questions:**

Can you provide results for longer time horizons to show if/when the coverage rate stabilizes?

Can you also discuss potential modifications to their algorithm that could accelerate convergence of the coverage rate, and any tradeoffs these modifications might introduce?

---

> ### Comment · Reviewer_TNwk · 2024-11-18
> **Where is author's response?**
>
> I am surprised that questions are not answered from the authors.

---

> ### Author Response · Authors · 2024-11-19
>
> Thank you for the thoughtful comments. We address your concerns below:
>
> - **Convergence Rate:**
>   There is no fundamental way to accelerate the convergence rate, as it is determined by the DKW inequality, which is tight in general. If smaller prediction sets are preferred, one can always select a lower coverage probability $\alpha$. We run our algorithm on ImageNet for 200K steps and present the results in the following table:
>
> | Step            | 1    | 28572 | 57143 | 85714 | 114286 | 142857 | 171428 | 200000 |
> |------------------|-------|-------|-------|-------|--------|--------|--------|--------|
> | Average Coverage | 1.00  | 0.94  | 0.93  | 0.92  | 0.92   | 0.92   | 0.92   | 0.91   |

---

> > ### Comment · Reviewer_TNwk · 2024-11-25
> >
> > Thank you for authors' response.
> >
> > One concern I have is the connection between your method and the literature on online conformal prediction. Specifically, the literature on adaptive conformal inference typically provides valid average coverage guarantees under any distribution shifts. However, if I understand correctly, your approach seems more aligned with the i.i.d. case with missing responses, where coverage might fail. The discussion of related literature in this aspect is limited.
> >
> > Additionally, I find the theoretical contribution somewhat limited. While Theorem 1 provides a regret analysis, it does not convincingly demonstrate that the method achieves any form of optimality. I suggest expanding the discussion on the theoretical contributions and connecting them more thoroughly to related works. Although the authors claim novelty for this setting, it is crucial to address the optimality of the proposed methods. In this regard, I believe the paper would benefit from some revisions.

---

### Official Review · Reviewer_bdoY · 2024-11-02

**Soundness:** 2
**Presentation:** 2
**Contribution:** 2
**Rating:** 5
**Confidence:** 3

**Summary:**

This paper considers the conformal prediction problem in the online learning setting with semi-bandit feedback. The authors proposed a new algorithm targeted at this setting and prove that it obtains sublinear regret bound. Then, the authors conduct some experiments to validate the high efficiency of the proposed algorithm over existing approaches based on conformal prediction in the online learning framework.

**Strengths:**

1. This paper studies an interesting and timely topic on conformal prediction with online learning.
2. I did not check the appendix in detail, but I feel the theoretical results are proper.
3. Several real-world experiments are conducted to validate the high performance of the proposed method.

**Weaknesses:**

1. I feel the presentation of this paper can be improved, The current version is still hard to follow. For example:
1.1 More background information on conformal prediction should be provided, such as the studies on conformal prediction in statistics and its recent applications on the online learning problem.
1.2 There are many definitions/notations in your work, and it is better to recall their definitions sometimes. For instance, in the second paragraph of page 3, it is better to define $\tau$ and $\tau_t$ there and I feel they are not previously defined. And in the pseudocode it is better to define $s_t$ there. The current version is hard to read.

2. Running time of algorithms should be reported in the paper since conformal prediction may lead to heavy computation, e.g. providing runtime comparisons between their proposed method and the baselines for each of the experimental tasks.

3. It is better to give some more real-world applications under your problem setting: a combination of conformal prediction and semi-bandit framework.

**Questions:**

1. Are your assumptions made in the paper common in the conformal prediction setting and the semi-bandit setting?
2. What is the theoretical novelty of your paper compared with the existing work? The analysis seems not very difficult for me. It is better to summarize the theoretical contributions in a small paragraph since I feel most techniques or results are similar to existing work under your assumptions.

---

> ### Author Response · Authors · 2024-11-19
>
> Thank you for the thoughtful comments. We address your concerns below:
>
> - **Assumptions:**
>   Both Assumption 2.1 and 2.2 are standard in regret analyses within the bandit literature (e.g., [1] and [2]).
>
>   Next, we can remove Assumption 2.3 with some extra work. First, we redefine $\tau^*$ as:
>   $$
>   \tau^* = \sup\\{\tau: G^*(\tau) \leq 1 - \alpha\\}.
>   $$
>   Furthermore, define $\alpha' = G^*(\tau^*)$. Now, the proof proceeds almost the same as before with $\alpha'$ substituted as $\alpha$. The only issue is that $\tau_t$ is computed using $\alpha$. However, because the DKW inequality applies to arbitrary CDFs, we can still show that $\tau_t \leq \tau^*$ with probability at least $1/T^2$. As a consequence, we can show that $G(\tau_t) \to 1 - \alpha'$, i.e., our algorithm maintains and converges to an $\alpha'$ coverage rate. With these modifications, our regret guarantee goes through as before. We are happy to update our paper with this relaxation.
>
>   [1] Kleinberg, Robert, Aleksandrs Slivkins, and Eli Upfal. "Multi-armed bandits in metric spaces." In *Proceedings of the fortieth annual ACM symposium on Theory of computing*, pp. 681-690. 2008.
>
>   [2] Auer, P. "Finite-time Analysis of the Multiarmed Bandit Problem." (2002).
>
> - **Theoretical Contributions:**
>   Our main theoretical contribution is to formalize and solve the stochastic online conformal prediction problem with semi-bandit feedback. To the best of our knowledge, our strategy for handling the truncated CDF is novel.
>
> - **Presentation of the Paper:**
>   We appreciate the reviewer's suggestion. Due to space constraints, most of the theoretical results have been deferred to the appendix. However, we will improve the organization of the paper in the next draft.

---

> > ### Comment · Reviewer_bdoY · 2024-11-25
> > **Thank you for your responses**
> >
> > I would like to thank the authors for your responses. After reading the rebuttal along with other reviewers' comments, I think the writing of this work can still be drastically improved as all the reviewers pointed out the writing is hard to follow. And a detailed comparison with existing works to showcase your theoretical contributions is also necessary. Furthermore, my concerns on the running time as well as the intuition/applications of this problem setting has not been addressed. I will keep my rating now. And the paper would benefit from a round of revision. Thank you again for your responses.

---

### Official Review · Reviewer_TUyt · 2024-11-03

**Soundness:** 3
**Presentation:** 2
**Contribution:** 2
**Rating:** 5
**Confidence:** 3

**Summary:**

The paper proposes an online conformal prediction method with semi-bandit feedback. The algorithm is based on an upper estimate of the score function's CDF, with sublinear regret bound under certain loss functions. Several experiments are conducted to show the effectiveness of the algorithm.

**Strengths:**

The algorithm enjoys a sublinear regret bound. It constructs compact prediction sets, while ensuring high coverage probability. The experimental results show significant improvement over the other baselines.

**Weaknesses:**

1. The theoretical result seems not very sufficient. From my understanding, coverage rate and the size of the prediction set are the two key metrics of practical interest. However, it is unclear how the result in Theorem 3.1 relates to these two metrics.
2. The sizes of the prediction sets are not reflected in the experimental results either.
3. The presentation quality can be improved. For example, in Fig. 1, error bars can be added, and I only observe 3 curves while 4 algorithms are in the legend. There are also some typos. For example, in line 429, "our our".

**Questions:**

1. Can the framework be extended to non-i.i.d setting? Such as black-box optimization, where the optimal x is searched for.
2. In assumption 2.1, the definition of loss is a bit abstract. Can the authors elaborate a bit more on how to interpret this loss in different practical applications?

---

> ### Author Response · Authors · 2024-11-19
>
> **Response to Reviewer 2:**
>
> Thank you for the thoughtful comments. We address the listed concerns below:
>
> - **Regret, Coverage Rate, and Prediction Set Size:**
>   Our regret guarantee ensures that our algorithm achieves the desired coverage rate and that its prediction set size converges to the size of the optimal prediction set. As explained in Section 3, we can recover the optimal prediction set by choosing $\tau_t = \tau^*$ for any $t$ if the true distribution $G^*$ is known. Since $G^*$ is unknown, we estimate it using our DKW inequality upper bound $\overline{G}_t$ and select its $1-\alpha$ cutoff as $\tau_t$. Note that we achieve sub-linear regret if and only if our estimates $\tau_t$ converge in probability to $\tau^*$. Consequently, we guarantee learning the optimal prediction set over time.
>
> - **Extension to Non-i.i.d. Settings:**
>   Extensions to the (adversarial) online learning setting are challenging due to the lack of an extension of the DKW inequality to this setting. For instance, learning thresholds is a closely related problem to ours, and the Littlestone dimension of thresholds is infinite (finite Littlestone dimension is a standard way to provide bounds in the online learning setting). Significantly different techniques would be needed, which we leave to future work.
>
> - **Interpretation of the Loss Function:**
>   The function $\phi$ represents the loss incurred when selecting a particular cutoff $\tau$. The bounded reward assumption in Assumption 2.2 typically holds in practice. For example, if $\phi$ corresponds to the prediction set size, the assumption holds because prediction set sizes are generally bounded. Alternatively, if $\phi$ measures the error of $\tau$ relative to $\tau^*$, then $\tau$ lies within a bounded range determined by the smallest possible score $f(x,y^*)$ (which is typically $\geq 0$ if $f(x,y^*)$ represents a probability).

---

> ### Comment · Reviewer_TUyt · 2024-11-25
> **Thanks for the response**
>
> I would like to thank the authors for the detailed responses. They help me better understand your paper. However, as also pointed out by other reviewers, the writing quality, including the experimental part, can be improved. Hence, I tend to not raise my rating.

---

### Official Review · Reviewer_Gwvt · 2024-11-04

**Soundness:** 3
**Presentation:** 2
**Contribution:** 2
**Rating:** 5
**Confidence:** 4

**Summary:**

- The paper studies the problem of conformal prediction in an online setting with semi-bandit feedback. The semi-bandit feedback is conditioned on whether or not the true label was present in the confidence set, if it is the true label is given back (and used for computing the score) for further refinement otherwise we only recieve that the true label is not present.
- The main contribution of this paper is an algorithm which performs online conformal prediciton, which the authors show achieves sub-linear regret upto a constant factor.
- The authors use the DFW inequality to construct upper bounds for the empirical CDF of the coverage parameter and adaptively update this upper bound using the semi bandit feedback.
- The authors show experiments on three different tasks including an image classification task, a document retrieval task and a second price auction task for three key metrics regret, coverage rate, undercoverage count. The tasks are benchmarked with 2 techniques: ACI and DLR.

**Strengths:**

1. To the best of my knowledge, the paper is the first to deal with the problem of online conformal prediction with semi bandit feedback. Therefore there is originality to the work.

2. The quality of some aspects of the paper is decent, especially the mathematical notation used is clean and easy to understand. The coverage of conformal prediction is sufficient. The introduction section is also well-written and provides intuitive exposure to conformal prediction.

3. The applications of the paper, especially in document retrieval, can be significant in a contemporary interactive document setting and even the annotation example for the image classification setting could be potentially useful in an active learning setting. Conformal prediction is also useful in creating prediction sets for black box models in general, and such a problem set has sufficient motivation in general.

4. The three metrics in the results are complementary to each other and match well with the claims of the paper.

**Weaknesses:**

1. The writing, in general, is a bit unclear and sloppy. Here a few examples:
- a. What is the source of randomness in the two levels of probability decomposition (The pac and the coverage)?
- b. The proof of theorem 1 can use some intuition
- c. The description of the Experiment is not well-written, for example, the desired coverage rate or other parameters? Does the paper use all the samples from imagenet or is their a subsampling?
- d. The theorem is not self-complete, everything referenced in the theorem should be defined the assumptions should be mentioned,
- e. The full form of PID and ACI is not mentioned in their first reference.
2. Assumption 2.2 might not hold in practice, especially in an active learning setting, where the loss function might be ill-defined for the first few rounds when the model lacks confidence.
3. In an active learning setting, do you ideally want a coverage set to decrease with time? Also in general, in active learning there is a constraint on the size of the samples that can be annotated. How much effort will be spent in identifying if the true label is there or not?
4.  The PAC bound does not reappear again in the paper or the theoretical results, what is the point of presenting it? Usually, there is a relation between the number of samples and the error probability, what is the relation in this case?
5. There are no margin-of-errors mentioned in the experimental results, therefore, it is difficult to establish the significance of these results. Also, the ACI line on cumulative regret is not discernible.

**Questions:**

1. What happens to the algorithm in practice if assumptions 2 and 3 are unsatisfied? Where does the regret analysis fail?

2. “As a consequence, our algorithm converges to the true \tau^*” Is the convergence a consequence of tau being a monotonic increasing sequence?

3. Are there changes to the aci algorithm that can achieve the same thing? What is the advantage of giving \tau_t as a feedback?

4. Why does the coverage rate increase and then decrease for ACI?

---

> ### Author Response · Authors · 2024-11-19
>
> Thank you for the thoughtful comments. We address the listed concerns below:
>
> - **Source of Randomness:**
>   Note that $\mathbb{P}_{(x,y^*)\sim D}(y^* \in C_Z(x)) \geq \alpha$ indicates that for any i.i.d. point sampled from $D$, the prediction set constructed using the cutoff $Z$ contains the ground-truth label with probability at least $\alpha$. The randomness originates from the test point sampling distribution, not from the choice of the cutoff $Z$. A second layer of randomness comes from the cutoff $Z$ itself, which is a random variable derived from the calibration dataset.
>
> - **Assumptions 2.2 and 2.3:**
>   The bounded reward assumption in Assumption 2.2 typically holds in practice. For example, if $\phi$ represents the prediction set size, this assumption holds because prediction set sizes are generally bounded. Alternatively, if $\phi$ measures the error of $\tau$ compared to $\tau^*$, then $\tau$ lies within a bounded range depending on the smallest possible score $f(x,y^*)$ (which is typically $\geq 0$ if $f(x,y^*)$ represents a probability).
>
>   Next, we can remove Assumption 2.3 with some extra work. First, we redefine $\tau^*$ as:
>   $$
>   \tau^* = \sup\\{\tau: G^*(\tau) \leq 1 - \alpha\\}.
>   $$
>   Furthermore, define $\alpha' = G^*(\tau^*)$. Now, the proof proceeds almost the same as before with $\alpha'$ substituted as $\alpha$. The only issue is that $\tau_t$ is computed using $\alpha$. However, because the DKW inequality applies to arbitrary CDFs, we can still show that $\tau_t \leq \tau^*$ with probability at least $1/T^2$. As a consequence, we can show that $G(\tau_t) \to 1 - \alpha'$, i.e., our algorithm maintains and converges to an $\alpha'$ coverage rate. With these modifications, our regret guarantee goes through as before. We are happy to update our paper with this relaxation.
>
> - **Active Learning and Coverage Rate:**
>   On average, the size of our coverage set decreases over time because the chosen cutoff $\tau_t$ increases with the number of observed samples. In our framework, one can always select a lower coverage threshold $\alpha$ if annotation cost is a primary concern. However, explicitly incorporating a constraint on prediction set size would be an interesting direction for future work.
>
> - **PAC Bound and Our Results:**
>   As we briefly mentioned in the introduction, our theoretical guarantee is stronger than the marginal guarantee typically seen in the conformal prediction literature. Specifically, our algorithm ensures the $\alpha$ coverage guarantee *for every $t$ with probability at least $1 - \delta$* over the entire horizon. In contrast, the marginal guarantee only maintains $\alpha$ coverage *on average* over the horizon. This means that algorithms relying solely on marginal guarantees may fail to maintain coverage for an arbitrary number of steps.
>
> - **Convergence of $\tau_t$:**
>   Our convergence guarantee is based on the empirical CDF converging to the true CDF (i.e., $\overline{G}_t$ converges to $G^*$ in probability).
>
> - **Modification of ACI:**
>   We believe there is no straightforward way to modify ACI to achieve our results. Furthermore, even in scenarios favorable to ACI, it can only provide a marginal guarantee, which is weaker than our guarantee, which is comparable to PAC.
>
> - **Coverage Rate of ACI:**
>   We believe the reason ACI's coverage rate initially increases and then decreases is that its estimate of the underlying CDF becomes increasingly biased over time due to the semi-bandit feedback. Conducting gradient descent in the quantile space is insufficient to correct this bias.

---

> > ### Comment · Reviewer_Gwvt · 2024-11-21
> > **Not Convinced with the author responses**
> >
> > I would request the authors to kindly expand the last 5 points of their response as the answers don't contain the context of the question and its difficulty to objectively judge them. Also, the reviewer would appreciate a point by point rebuttal, some of the concerns were not addressed.
> > Also, it would be great if the authors have a revision which incorporates some of the changes with regard to the clarity in the paper and other related concerns. Also, any clarification regarding the margin of errors would be appreciated.

---

### Meta-Review · Area_Chair_Z9Lm · 2024-12-11

**Metareview:**

This paper introduces a novel online conformal prediction algorithm with semi-bandit feedback, achieving sublinear regret and demonstrating its applicability across tasks like image classification, document retrieval, and auction price-setting. The main contribution is the development of a regret bound for this new setting.

Reviewers praised the paper's novelty, but raised concerns about its clarity, noting difficulties in following the arguments and defining key notations. The theoretical contributions were considered limited, with a need for more comparison to existing work. Additionally, experimental details such as error margins, running time comparisons, and longer-term results were missing.

**Additional Comments On Reviewer Discussion:**

Overall, while the paper presents an interesting and novel approach, the reviewers agree that it requires substantial revisions in terms of clarity, theoretical depth, and experimental detail before it can be considered for acceptance.

---

### Decision · Program_Chairs · 2025-01-22

Reject